# Molecular Mechanisms of Kidney Injury and Repair in Arterial Hypertension

**DOI:** 10.3390/ijms20092138

**Published:** 2019-04-30

**Authors:** Laura Katharina Sievers, Kai-Uwe Eckardt

**Affiliations:** 1Department of Nephrology and Medical Intensive Care, Charité-Universitätsmedizin Berlin, 13353 Berlin, Germany; kai-uwe.eckardt@charite.de; 2Max-Delbrück Center for Molecular Medicine in the Helmholtz Association, 13125 Berlin, Germany; 3Experimental and Clinical Research Center, a joint cooperation of Max-Delbrück Center for Molecular Medicine and Charité-Universitätsmedizin Berlin, 13125 Berlin, Germany; 4Berlin Institute of Health (BIH), 10178 Berlin, Germany

**Keywords:** hypertension, kidney, molecular signaling

## Abstract

The global burden of chronic kidney disease is rising. The etiologies, heterogeneous, and arterial hypertension, are key factors contributing to the development and progression of chronic kidney disease. Arterial hypertension is induced and maintained by a complex network of systemic signaling pathways, such as the hormonal axis of the renin-angiotensin-aldosterone system, hemodynamic alterations affecting blood flow, oxygen supply, and the immune system. This review summarizes the clinical and histopathological features of hypertensive kidney injury and focusses on the interplay of distinct systemic signaling pathways, which drive hypertensive kidney injury in distinct cell types of the kidney. There are several parallels between hypertension-induced molecular signaling cascades in the renal epithelial, endothelial, interstitial, and immune cells. Angiotensin II signaling via the AT1R, hypoxia induced HIFα activation and mechanotransduction are closely interacting and further triggering the adaptions of metabolism, cytoskeletal rearrangement, and profibrotic TGF signaling. The interplay of these, and other cellular pathways, is crucial to balancing the injury and repair of the kidneys and determines the progression of hypertensive kidney disease.

## 1. Introduction

Arterial hypertension has a large prevalence in the general population and is associated with a wide range of cardiovascular complications and chronic damage to the heart, brain, vasculature, eyes, kidney, and other organs. Hypertensive nephropathy is regarded as the second leading cause of end-stage renal disease (ESRD), outnumbered only by diabetic nephropathy. However, in many cases, it is hard to determine the primary underlying cause of chronic kidney disease (CKD). Arterial hypertension is a typical complication of CKD, irrespective of its etiology, and it is often difficult to differentiate whether increased blood pressure is the cause and/or consequence of impaired kidney function. In any case, coexisting arterial hypertension accelerates the progression of CKD and increases the cardiovascular risk in CKD patients [1]. The clinical course and histopathological characteristics of kidney injury in hypertensive kidney disease may vary, not only on comorbidities, but also in environmental factors and genetic predisposition. For example, patients of black ethnicity are at a much higher risk for rapidly progressing CKD. Given the importance of hypertension for the course of kidney disease, a thorough understanding of the molecular mechanisms of kidney injury and repair, in arterial hypertension, appears fundamental to the development of novel approaches against the progression of CKD worldwide. With the increasing availability of novel therapeutic strategies targeting molecular pathways, a classification of CKD, based on the predominant molecular pathology, rather than clinical correlations/etiologies might become a promising approach in the future.

Arterial hypertension is caused and sustained by a complex network of systemic signaling pathways. The renin-angiotensin-aldosterone-system (RAAS) is one important hormonal axis in hypertension. In addition, several other hormones, such as corticosteroids, catecholamines, thyroid hormones, sex hormones, and others contribute to the regulation of blood pressure. Furthermore, it has been shown that the immune system plays an important role in the development and maintenance of hypertension. For example, the balance between IL17 producing T lymphocytes (Th17) and regulatory T cells (Treg) is dysequilibrated in hypertensive patients, favoring Th17 cells. Other conditions favoring Th17 cells can drive hypertension. In this review we aim to provide a brief overview of clinical and histopathological characteristics of kidney injury in hypertensive kidney disease, and of the systemic signaling pathways and important aspects of the immune system, in arterial hypertension. Moreover, we will describe molecular mechanisms in hypertensive kidney injury and repair, including angiotensin II (Ang II) signaling in different cell types, hypoxia signaling, distinct pro-inflammatory pathways, and TGF-associated profibrotic signaling.

## 2. Clinical and Histopathological Characteristics of Kidney Injury in Arterial Hypertension

“Hypertensive nephropathy”, also known as “hypertensive nephrosclerosis”, is traditionally characterized by a combination of pathological changes of the pre- and intra-glomerular microvasculature and the tubulointerstitium. The histopathology can hardly distinguish whether arterial hypertension is the primary cause of kidney dysfunction or whether increased blood pressure occurs as a comorbidity, which drives CKD progression. Thus, the term hypertensive nephropathy summarizes both conditions. The severity of blood pressure elevation often correlates with the degree of renal damage. In many cases, hypertensive nephrosclerosis shows a slow progression, which is historically classified as “benign nephrosclerosis”. In contrast, accelerated nephrosclerosis, histopathologically characterized by fibrinoid necrosis and/or myointimal cell proliferation, is classified as “malignant nephrosclerosis” and frequently leads to ESRD [2]. Hypertension-induced kidney damage involves different cell types and anatomical structures in the kidney, including the vasculature, glomeruli, tubulointerstitium, and immune cells. The muscular arteries and arterioles of the kidney parenchyma show progressive intimal thickening during aging, but besides age this process correlates with arterial hypertension [3]. The thickening is caused by collagen deposition and spreading of elastic fibers and myofibroblasts, and ultimately leads to more pulsatile blood flow in kidney arterioles [3]. Another histopathological characteristic of hypertensive nephrosclerosis is arteriosclerosis of the afferent arterioles, also referred to as “afferent arteriolar hyalinosis”. These typical hyaline deposits are the consequence of a pathogenetic cascade of atrophy of vascular smooth muscle cells, increased endothelial leakiness and plasma protein extravasation, leading to sub-endothelial protein accumulation [3]. Although this process is associated with hypertension, to some extent, it occurs in all aging kidneys [3]. The glomerular involvement is heterogeneous; there is a side-by-side normal morphology, ischemic, obliterated glomeruli, with collapsed capillaries and focal-segmental-glomerular sclerosis-(FSGS)-like lesions, with partially sclerotic glomerular adhesions.

Another hallmark of hypertensive kidney injury is tubular atrophy, accompanied by interstitial fibrosis. Following a loss of functional nephrons, the surviving nephrons initially maintain total kidney function, but the concomitant hemodynamic adaptations results in an oxygen supply-demand mismatch and ultimately tubulointerstitial hypoxia [4]. Tubulointerstitial hypoxia presumably contributes to the progression of tubular damage and renal functional impairment [5,6]. This relative hypoxia is further aggravated by impaired oxygen delivery to the kidneys during hypertension, due to vasoconstriction hormones, including components of the RAAS, prostaglandins, and endothelin [5]. During the course of CKD, the loss of peritubular capillaries further aggravates tubulointerstitial hypoxia and damage. Hypoxic conditions trigger mitochondrial dysfunction [4] and can activate the transcription factor hypoxia-inducible factor (HIF) [7]. The signaling pathways which promote a maladaptive phenotype at the cellular level will be discussed below. Besides the characteristic vascular adaptations and glomerular pathology, kidney histology of hypertensive nephrosclerosis may also show trans-differentiation and apoptosis of tubular cells, increased peritubular fibrosis, fibroblasts proliferation, and increased interstitial inflammation.

## 3. Systemic Signaling Pathways and the Immune System in Arterial Hypertension

For decades, the pathophysiology of arterial hypertension has primarily been attributed to vasoconstriction hormones, including RAAS (Figure 1), prostaglandins, and endothelin 1 (ET1). Among these hormonal axes, the RAAS has received particular attention, and was shown to have a crucial influence on blood pressure and target organ damage. This important signaling axis includes several components that are activated stepwise: First, the 452 amino acid peptide pro-hormone angiotensinogen, which is synthesized and excreted by the liver, is cleaved to the ten amino acid peptide Angiotensin I by the protease renin (Figure 1). Angiotensin (Ang) I is then cleaved to the eight amino acid peptide hormone, Ang II, by angiotensin converting enzyme (ACE) (Figure 1). Although, the pro-hormone Ang I and other metabolites of angiontensinogen, such as Ang 1–7, which result from cleavage by other proteases, have effects on the vascular endothelia and other tissues. Here, we will focus on the canonical RAAS signaling and biological effects of Ang II. Ang II binds to the Ang II-receptor 1 (AT1R) in different cell types [8]: In the vasculature, Ang II leads to vasoconstriction and in the adrenal cortex, it increases the secretion of aldosterone (Figure 1). In the kidney, Ang II leads to tubular retention of NaCl and water (Figure 1). Further, Ang II increases the sympathetic activity, which also elevates blood pressure (Figure 1). The systemic vascular tone results from the interplay of many different vasoconstricting and vasodilating stimuli. Within this complex interplay, the pathophysiological relevance of the RAAS for blood pressure regulation is underlined by the clinical significance of RAAS inhibition. State-of-the-art clinical management of arterial hypertension includes, pharmacological targeting of the RAAS with ACE-inhibitors, AT1R blockers, direct renin inhibitors, and aldosterone antagonists [9]. The important pathophysiological contribution and clinical relevance of the RAAS, beyond blood pressure control, is underlined by the fact that treatment with ACE-inhibitors or AT1R blockers shows significant clinical benefits for cardiovascular morbidity and mortality.

ET1 expression may be regulated by inflammatory mediators, shear stress, and glucose. ET1 has an ambiguous role on the vascular tone. Binding of ET1 to the ET1 receptor A (ETA) of vascular smooth muscle cells causes vasoconstriction, while binding to the ET1 receptor B (ETB) has an indirect vasodilating effect via increased release of nitric oxide (NO). ETA antagonists are clinically used in pulmonary arterial hypertension. Further, ETA antagonists may slow the progression of diabetic nephropathy, but until recently, ET1 antagonists are not licensed in arterial hypertension. Sympathetic activity may be induced by central nervous regulation associated with stress, pain, or cold, It is affected by the RAAS and modifies peripheral resistance. In the last years, several non-pharmacological approaches addressing the autonomic nervous system in arterial hypertension have been developed. Renal sympathetic denervation, carotid baroreflex activation therapy, central iliac arteriovenous anastomosis, carotid body ablation, median nerve stimulation, vagal nerve stimulation, and deep brain stimulation continue to be evaluated with respect to their capacity to lower blood pressure and blood pressure associated target organ damage [10].

Carotid baroreflex activation therapy resulted in a sustained reduction of arterial blood pressure in patients with resistant hypertension in the Rheos Pivotal Trial [11]. Since renal sympathetic over-activity is associated with the pathogenesis and progression of arterial hypertension and CKD, catheter-based renal denervation has been a promising approach in reducing blood pressure and attenuating renal functional decline in hypertensive CKD [12]. Further, meta-analyses have shown beneficial effects in patients with heart failure [12]. After promising animal studies and early trials in patients, subsequent clinical studies could not replicate the positive outcome [12,13,14]. There are ongoing trials, which aim to resolve the conflicting evidence and to provide data on the long-term safety and efficacy of renal denervation [15,16,17,18].

Besides the hormonal and autonomous axes, immune mechanisms play an important role in the pathogenesis of arterial hypertension by contributing to both, the development of hypertension and of hypertensive end-organ damage (Figure 2). CD4^+^ lymphocytes are key cell types involved in the balance between Treg and Th17 cells, which originate from naive CD4^+^ cells under specific skewing conditions. Several drivers of hypertension, including Ang II-induced signaling, dietary NaCl, the gut microbiome, at least partly exert their effect via immune mechanisms, and in particular, the regulation of the Treg/Th17 balance (Figure 2). Th17 cells are associated with hypertension and hypertensive end-organ damage, while a protective role is attributed to Treg. For example, it has been shown that Treg deficiency exaggerated Ang II-induced microvascular injury by enhancing immune responses [19]. Ang II was a driver of Treg infiltration in the aortic wall and renal cortex and, in a model of Treg deficient mice, the adoptive transfer of Treg cells prevented Ang II-induced hypertension and vascular injury [20]. Further, it has been demonstrated that Ang II increased the secretion of the pro- inflammatory cytokine IL17 by Th17 cells and that the activation Th17 cells in response to high salt was associated with accelerated fibrosis [21]. Notably, the cytokine IL17 has a differentiated function in diabetic kidney disease: Ablation of intrarenal Th17 cells ameliorated early diabetic nephropathy [22], while the administration of low-dose IL17A had beneficial effects [23]. The ambiguity of IL17 signaling in hypertensive kidney disease remains to be elucidated. Interestingly, the induction of Th17 cells seems to depend on gut microbiota, particularly salt-sensitive lactobacilli. Increased dietary salt depleted these microbiota, increased Th17 cells, and increased the blood pressure in mice and humans [24]. Besides salt intake, intestinal metabolites, such as the short-chain fatty acid propionate have quantitative effects on the systemic composition of immune cell subsets and blood pressure [25] (Figure 2). Intracellular pathways, that balance the differentiation of Treg and Th17 include. TGFβ, IL6, RORγT, Hippo/TAZ, and HIF-1. All these pathways are directly influenced by Ang II/AT1R signaling. Besides the prominent role of the Treg/Th17 cells in hypertensive end-organ damage, innate immune cells contribute to hypertensive mechanisms. Neutrophils, monocytes/macrophages, dendritic cells, myeloid-derived suppressor cells and innate lymphoid cells may trigger both, blood pressure elevation on the one hand, and profibrotic inflammatory processes in end-organs on the other hand [26]. Innate and adaptive immune mechanisms closely interact in the pathogenesis of hypertension and hypertensive organ damage. For example, monocyte/macrophages and γδ T cells seem to play a crucial role in the initiation of hypertension by priming adaptive immune cells, thereby triggering vascular inflammation and blood pressure elevation or, if their signaling limits the inflammatory response, protecting against vascular injury [26].

## 4. Molecular Mechanisms in Hypertensive Kidney Injury

Many different anatomical parts of the kidney, with distinct cell types, are affected by hypertensive kidney injury: Cells forming the nephron, including glomerular podocytes and tubular cells, the vasculature with endothelium and smooth muscle cells, interstital fibroblasts and resident, as well as circulating immune cells of adaptive and innate immune response, are involved (Figure 3). Although novel techniques, such as genome-wide association studies, single cell proteomics or deep sequencing are on the brink of their application in clinical studies, detailed reports scrutinizing cell-type specific molecular mechanisms of hypertensive kidney injury are still missing [27,28]. When subsequently discussing important molecular mechanisms in hypertensive kidney injury and repair, we will focus on the molecular signaling pathways of Ang II signaling, hypoxia, pro- inflammatory signaling, TGF, and profibrotic pathways. Although, the net effect of mechanisms promoting injury and repair is highly cell type- and context-dependent, there are several parallels with regard to the involved molecular signaling pathways across different cell types (Figure 3). 

The peptide hormone Ang II is an important mediator in the pathophysiology of arterial hypertension, hypertensive end-organ damage, and also hypertensive kidney injury. The AT1R is expressed in various cell types, including podocytes, renal tubular cells, and immune cells. Further, Ang II exerts indirect effects via changes of the glomerular hemodynamics, leading to increased filtration pressure, increased filtration of NaCl and consecutive adaptations of the tubular salt and water handling. Furthermore, Ang II from the systemic circulation increases the tubular expression of angiotensinogen, ACE, renin and AT1R [29]. This feed-forward loop mechanism is at least partly driven by cytokines produced by activated T lymphocytes such as IL6 and IFNγ [30]. Consistent with the prominent glomerular histopathology and the clinical presentation of albuminuria in hypertensive kidney injury, elevated levels of Ang II lead to severe injury of glomerular podocytes with foot process effacement and ultimately podocyte loss. Podocyte depletion is an irreversible hallmark of glomerular injury in CKD, and the beneficial effect of Ang II blockade on the progression of hypertensive CKD correlates with attenuated podocyte loss [31]. Ang II causes podocyte injury via several pathways. Binding of Ang II to the G-protein-coupled AT1R activates an intracellular cascade of kinases, including the activation of phospholipase C, calcium-calmodulin binding, phosphorylation of the protein kinases extracellular-signal regulated kinases (ERK), ribosomal S6 kinase (RSK), protein kinase C (PKC), protein kinase A (PKA), and the activation of adenylate cyclase leading to the generation of cyclic AMP. Already decades ago, it has been demonstrated that a sustained intracellular rise in calcium, as induced by Ang II, has detrimental effects on podocyte viability [32,33]. The calcium influx may be aggravated by simultaneously increased expression of calcium channels, such as TRPC6 [34]. Further, Ang II signaling in podocytes is closely linked to the integrity of the actin cytoskeleton. It has been shown that short term treatment with Ang II regulates the phosphorylation of various actin-associated proteins in podocytes and that e.g., the increased phosphorylation of lymphocyte cytosolic protein 1 (LCP1) leads to alterations in actin bundling, increased filopodia and lamellipodia formation, and membrane ruffling [35]. Besides the regulation of intracellular cytoskeletal components, Ang II has a direct impact on nephrin, a fundamental extracellular adaptor protein crosslinking podocyte at the slit diaphragm. In models of Ang II induced hypertension and kidney injury, nephrin is dephosphorylated in a caveolin-1- [36] and c-Abl [37]-dependent way, ultimately leading to the disintegration of the slit diaphragm. Interestingly, it has been shown that Ang II also regulates the Hippo pathway, a conserved signaling pathway controlling cell proliferation and cell death via a kinase cascade, that regulates the nuclear shuttling of the transcription factors YAP and TAZ. Ang II inactivated the Hippo pathway by decreasing the activity of LATS kinase, consequently increasing the nuclear abundance of the transcription factor YAP in Human Embryonic Kidney cells [38] and TAZ is activated in the tubulointerstitium in mouse models of kidney injury [39]. At the same time, Ang II effects on YAP/TAZ were absent in podocytes, which are postmitotic cells with low baseline Hippo pathway activity [38]. Further, Ang II/AT1R signaling modifies the lipid metabolism and induced lipid droplet accumulation and expression of lipid droplet marker protein in podocytes [40]. The effects of Ang II on renal cells have been extensively studied, but the interrelation of different signaling networks is still not completely understood. Further, Ang II affects the infiltration of immune cells. AT1R signaling drives the differentiation of CD4 lymphocytes to Th17 cells: Silencing of AT1R with siRNA reduced the fraction of Th17 polarized cells and IL17 expression while Ang II treatment increased the secretion of the pro-inflammatory cytokine IL17 by Th17 [21] wheras, in the presence of ACE inhibition with lisinopril, Foxp3 positive Treg were enhanced [41]. Besides the significance of Th17/Treg cells for the development and progression of arterial hypertension and hypertensive end-organ damage, γδ T-cells contribute to Ang II-induced elevation of blood pressure, endothelial dysfunction, and the activation of innate and adaptive immune response [42]. Further, Ang II increased monocyte chemotactic protein-1 expression in the vasculature, accompanied by higher monocyte/macrophage infiltration and pro-inflammatory macrophage polarization in the renal cortex [19]. Ang II also acts as a pro-fibrogenic cytokine regulating renal cell proliferation, synthesis and degradation of the extracellular matrix by various pathways, e.g., TGFβ [43].

TGFβ is secreted by many different cell types, including monocytes/macrophages, and a key function is the regulation of inflammatory processes and profibrotic signaling. Increased systemic levels of TGFβ are associated with faster CKD progression and TGFβ polymorphisms have been identified as risk factors for ESRD [44,45]. However, the effect of TGFβ is highly context-dependent. Ang II induces the expression of the TGFβ downstream target connective tissue growth factor β (CTGF), which is associated with endothelial mesenchymal transition. The profibrotic action of Ang II may be aggravated by interaction with other pathways such as hypoxia, the activation of plasminogen activator inhibitor-1, or ET1 signaling. Interestingly, the Hippo pathway transcription factor, TAZ, was activated in the renal tubulointerstitium in different models of kidney injury. Nuclear accumulation of TAZ depended on TGFβ1-signaling and went along with fibrosis progression [39]. M2 macrophage polarization, which significantly correlates with kidney fibrosis, is driven by TGFβ1-induced activation of YAP/TAZ, too [46]. The action of bone morphogenetic protein (BMP), which also belongs to the TGF protein family, in the kidney is not completely understood. In healthy renal tubules, BMP is constitutively active and its reactivation, after an injury that suppressed BMP expression, correlates with functional recovery [47].

Ang II, ET1 and catecholamines both exert their function via G-protein coupled receptors (GPCRs), such as the AT1R, ETA, ETA and b-adrenergic receptors. Current pharmacological approaches mainly target these GPCRs. The sensitivity of hormone-induced GPCR activation is modified by G-protein coupled receptor kinases (GRKs), and the regulator of G-Protein Signaling proteins [48]. It has been demonstrated that the dysregulation of the GRK isoforms 2–6 correlates with increased blood pressure, poor response to antihypertensive treatment, and adverse cardiovascular outcomes [49]. Renal fibrosis could be alleviated by the inhibition of the G-protein βγ-subunit in mouse models of cardiorenal syndrome and ischemic acute kidney injury [50]. Further, inhibition of the downstream kinase PKA may reduce profibrotic signaling and profibrogenic epigenetic priming in diabetic kidney disease [51]. Pharmacological modification of GPCR downstream targets may be a promising approach in hypertensive CKD.

Another important mechanism, contributing to hypertensive nephrosclerosis, is hypoxia (Figure 3). Chronic ischemic tubulointerstitial damage, caused by altered hemodynamics, increased oxygen demand, and loss of peritubular capillaries is a hallmark of progressive CKD. Under hypoxic conditions, HIF, and its oxygen-regulated isoforms, HIF1α and HIF2α, are stabilized and promote cellular adaptation to the decreased oxygen supply and influence cell proliferation, survival and metabolism. The increased expression of HIF1α was reported to correlate with glomerular injury and promote hypertensive CKD [52]. HIF1α gene expression in renal endothelia was induced by Ang II in a Nuclear Factor-κB (NFκB)-dependent manner [52]. It has also been shown, that reciprocal positive transcriptional regulation leads to persistent activation of HIF1α and NFκB genes and drives disease progression [52]. The profibrotic action of HIF1α is partly mediated via induction of TGFβ and its proinflammatory downstream target CTGF [53]. Interestingly, HIF2α, which is primarily expressed in non-epithelial cells is associated with ambiguous effects on renal fibrosis: In early stages of CKD, activation of HIF2α worsened renal fibrosis but did not lead to renal functional impairment [54]. In another study, overexpression of HIF2α was sufficient to induce kidney fibrosis [55]. At later stages of CKD, HIF2α activation, in part, activated typical hypoxia-induced target genes of HIF1α such as VEGF, fibronectin, and type 1 collagen but restored the renal vasculature and thereby ameliorated renal dysfunction and fibrosis [54].

Besides the development of hypoxia, the hemodynamic changes associated with a defective renal autoregulation of blood pressure, during hypertensive nephropathy, and other types of chronic kidney disease, are associated with flow- and shear-stress dependent signaling in endothelial cells (Figure 3). In response to increased blood flow, endothelial cells release ATP, which in turn activates endothelial cells or adjacent immune cells, in a paracrine fashion via trinucleotide-receptors, from the ionotropic P2X and metabotropic P2Y receptor family. For example, hypertension correlates with overexpression and activation of the P2 × 7, P2Y12, and P2X1 receptors [56]. The blockade of these receptors inhibits renal vasoconstriction induced by Ang II, restores microvascular dysfunction, and improves the Ang II-associated regional hypoxia [56,57]. In addition to the microvascular implications, the release of inflammatory mediators, such as interleukins in the tubulointerstitium seems to be related to the activation of P2X/P2Y receptors [56]. Secondly, the increased filtration pressure may lead to higher urinary flow and flow-activated signaling pathways in tubular cells, which may lose epithelial characteristics, such as the in the expression of cell junctional proteins [58]. CKD goes along with secretion of pro-inflammatory cytokines by activated immune cells, interstitial cells, and tubular epithelial cells [59]. For example, hypertensive kidney injury correlates with phosphorylation of the cytoskeleton-associated protein cofilin 1, formation of actin stress fibers, nuclear translocation of NFκB and expression of downstream inflammatory factors in renal tubular epithelial cells. [59]. 

Further, infiltration of IL17 secreting T lymphocytes may lead to vascular dysfunction. IL17 acts on vascular smooth muscle cells and perivascular fibroblasts, in order to increase reactive oxygen species (ROS) production, collagen synthesis, and chemokine production. In parallel, IL 17 reduces the bioavailability of NO and, vasodilatation, enhances vascular stiffness and the recruitment of immune cells [30].

## 5. Approaches to Foster Regeneration and Repair in Hypertensive Nephropathy

Despite recent advances in the understanding of molecular mechanisms, contributing to hypertensive nephropathy, the interplay between the different signaling networks in each cell type and—even more complex—between distinct cell types, is still incompletely deciphered. However, “omics” approaches such as single-cell proteomics and metabolomics, in conjunction with advanced imaging techniques, comprise excellent opportunities to gain further insight into this complex signaling network. So far, clinical management of hypertensive CKD focusses on the deceleration of CKD progression. A major aim is to stop all comorbidities, which are known to drive the progression of hypertensive CKD: Blood pressure should ideally be normalized, blood glucose control should be excellent, dietary salt may be restricted, and nephrotoxic medications are to be avoided. Further, pharmacological Ang II blockade has beneficial effects on the progression of hypertensive end-organ damage. Therapeutic approaches to other signaling pathways, such as HIF stabilization, ET1 blockade, anti-inflammatory therapy or immunosuppression, have so far either not been tested or not proven safe and efficient. In pre-clinical research, stem cell based approaches are evaluated to improve kidney regeneration in CKD. However, in nephrology in general, and especially with regard to hypertensive nephropathy, regenerative medicine techniques are still in their children’s shoes. Studies are carried out with embryonic stem cells, mesenchymal stem cells, adipose stem cells, amniotic fluid stem cells, and renal progenitors, but also pluripotent cells. So far, the results from these studies are conflicting. However, even if stem cells and multipotent cells might not directly differentiate and replace damaged cells, they might produce protective and regenerative factors, and thereby enable functional improvement [60,61]. 

The bench-to-bedside transition of insight into the molecular mechanisms of CKD in hypertensive patients, and also a profound understanding of the molecular pathogenesis of CKD with other/mixed etiology, may lead to the development novel treatment approaches. Classifying CKD, based on the individual predominant molecular pathology, rather than traditional clinical etiologies would be an important step in developing focused molecular treatments. There is still a long way ahead, but once achieved, personalized medicine holds promise in CKD.

## Figures and Tables

**Figure 1 ijms-20-02138-f001:**
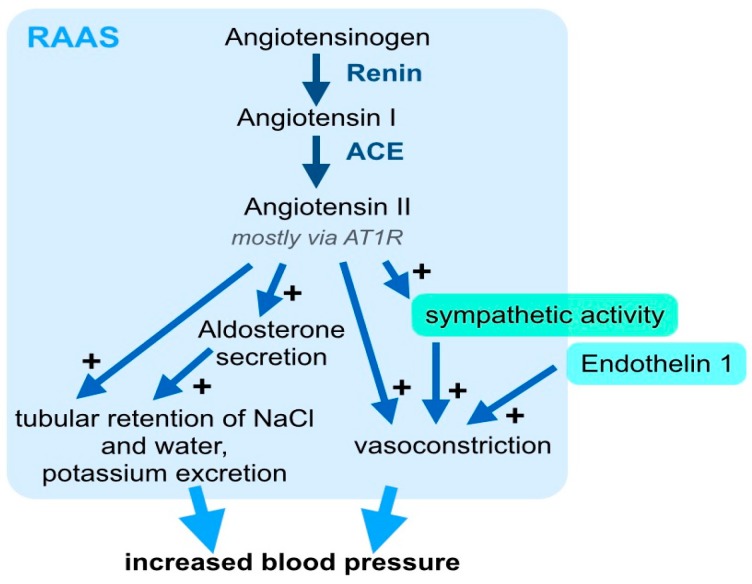
The renin-angiotensin-aldosterone system (RAAS) is a key hormonal axis in the pathogenesis of arterial hypertension. The RAAS combines a regulatory network, including angiotensin II (Ang II) and aldosterone, which increase systemic blood pressure through a concerted mechanism. Angiotensinogen is enzymatically cleaved by renin and angiotensin-converting enzyme (ACE) to generate Angiotensin II, which, mainly via the Ang II receptor 1 (AT1R), stimulates the secretion of aldosterone, acts as a vasoconstrictor and leads to renal tubular retention of NaCl and water. For simplification, the negative feedback loop, which physiologically limits an excess activation of these pathways, is not included in this graph. Besides Ang II, endothelin 1 and increased sympathetic activity, which may be induced by RAAS activation or independent stimuli, also contribute to systemic vasoconstriction and increased peripheral resistance.

**Figure 2 ijms-20-02138-f002:**
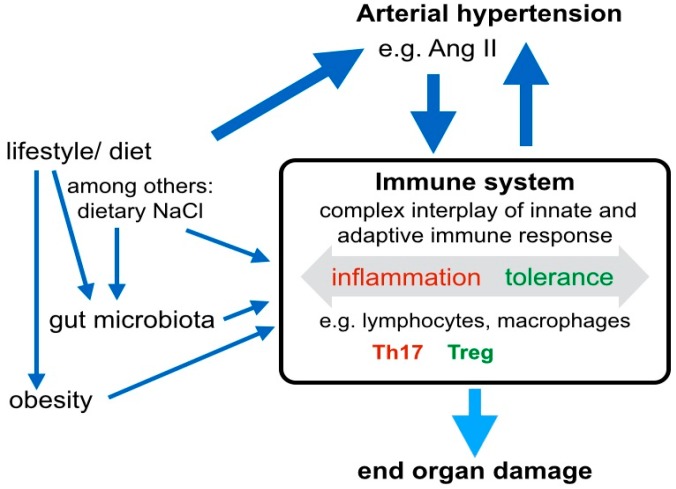
The immune system closely interacts with hormonal and environmental modifiers to control blood pressure. The immune system plays an important role in the pathogenesis of arterial hypertension and hypertensive end-organ damage. Lifestyle factors, such as the dietary nutrient composition and sodium intake directly, and via alterations in the gut microbiome, have influence on the cellular compositions of the immune system. Distinct immune cell subtypes, such as IL17 producing Th17 lymphocytes contribute to increased blood pressure. In parallel, existing arterial hypertension and Ang II favor Th17 polarization of T lymphocytes. Besides the effect of immune cells on blood pressure, the balance of various cells from the adaptive and innate immune system, including lymphocytes and macrophages is critical for end organ damage.

**Figure 3 ijms-20-02138-f003:**
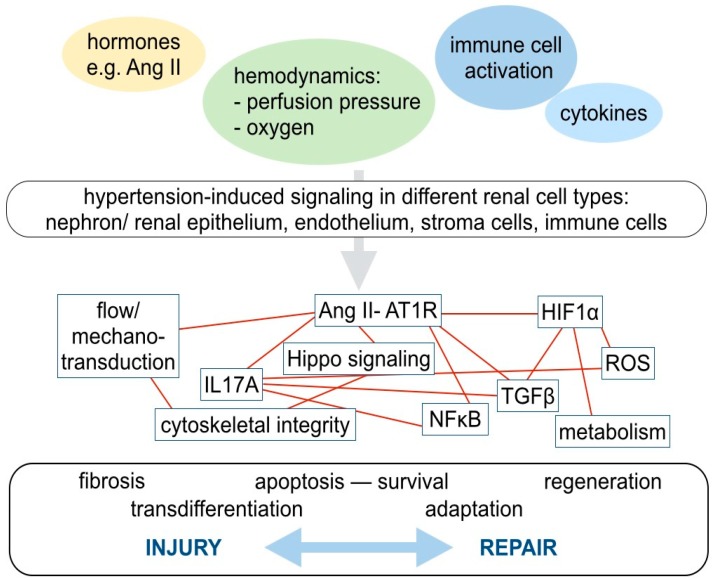
A complex network of various molecular pathways contributes to cellular adaptions, injury, and regenerative capacity in hypertensive kidney injury. In arterial hypertension, the cells in the kidney are exposed to various extracellular stimuli, including hormones, such as Ang II, altered hemodynamics with increased perfusion pressure, and potentially decreased oxygen supply and activated immune cells, which in turn secrete different cytokines. These stimuli affect all renal cell types: Nephron forming renal epithelial cells, endothelial cells, stroma, and immune cells. Although the net effect on injury and repair is highly cell type- and context-dependent, there are several parallels regarding the involved molecular signaling pathways. These pathways, or rather, signaling networks, are highly interrelated. For example, AT1R-mediated Ang II signaling, the Hippo pathway, pro- inflammatory signaling, including IL17 secretion and activation of TGFβ associated profibrotic signaling closely interact. Mechano-transduction and Hippo signaling, among others, influence the cytoskeletal integrity. Hypoxia-inducible factor (HIF) is stabilized by hypoxia, induces metabolic adaptions and may increase TGFβ- associated profibrotic pathways.

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
