# Peer review of "Molecular Mechanisms of Kidney Injury and Repair in Arterial Hypertension"

_ijms, 2019, doi:10.3390/ijms20092138_

Round 1

Reviewer 1 Report

This manuscript provides an overview of the molecular mechanisms involved in kidney injury and repair in arterial hypertension.conditions. The review is properly articulated by topics and sub-sections, and is properly written, providing a good overview of the role RAAS, immunity, endothelin 1 play in promoting injury and repair of the kidney in the context of the progression of hypertensive kidney disease. 

Minor: 

few typos need to be emended 

a reference needs to be entered at line 155.

Author Response

Dear reviewer,
thank you very much, we appreciate your valuable comments. The revised version of our manuscript has been improved according to the suggestions of three reviewers.
In particular, the description of pathomechanisms underlying hypertensive kidney injury has been extended and the manuscript has been proof-read carefully.
It has been clarified which pathways other than the RAAS contribute to immune cell stimulation, sympathetic nerve activation and endothelin-1 in the context of hypertensive CKD. Further, the discussion of GPCR signaling was extended and it was clarified, that based on data from models of diabetic nephropathy, IL17 might play an ambiguous role.
We hope to find your approval and contentment with our corrections.

Reviewer 2 Report

Overall a good review of the mechanisms for hypertensive kidney disease. I have the following suggestions/comments

1. The authors should acknowledge or address the fact that immune cell stimulation, sympathetic nerve activation etc are mostly described in the context of RAAS. What else stimulates these pathways

2. Isn't PKA activation via Gs beneficial?

3. The authors should comment on Mohamed et al  J Am Soc Nephrol 27, 745–765 (2016) 

4. Ln122 the authors probably mean AT1R blockers

5. Ln155 the authors reminder for a REF should be completed

6. Ln170 grammar and syntax need addressing, there are few other instances in the manuscript as well

7. Ln361 grammar correction

Author Response

Dear reviewer,
thank you very much, we appreciate your valuable comments. The revised version of our manuscript has been improved according to your suggestions.
In particular, the description of pathomechanisms underlying hypertensive kidney injury has been extended and the manuscript has been proof-read carefully.
Point-to-point reply to your commentaries:

Comment 1:
The authors should acknowledge or address the fact that immune cell stimulation, sympathetic nerve activation etc are mostly described in the context of RAAS. What else stimulates these pathways
Answer:
Thank you very much for this suggestion. It has been specified in the text which pathways other than the RAAS contribute to immune cell stimulation, sympathetic nerve activation and endothelin-1 in the context of hypertensive CKD. The caption of Figure 1 was adapted to „Besides Ang II, endothelin 1 and an increased sympathetic activity, which may be induced by RAAS activation or independent stimuli, also contribute to systemic vasoconstriction and increased peripheral resistance.“ and the text   page 3-4 was revised to „ET1 expression may be regulated by inflammatory mediators, shear stress and glucose for example. ET1 has an ambiguous role on the vascular tone. […] Further, sympathetic activity, which may be induced by central nervous regulation associated to stress, pain or cold but which is also affected by the RAAS, modifies the peripheral resistance.“

Comment 2:
Isn't PKA activation via Gs beneficial?
Answer:
PKA activation via GPCRs has ambiguous functions in different cell types. In podocytes for example, PKA activation may contribute to the adaptation to harmful external stimuli while in many other models, e.g. diabetic nephropathy, it has deleterious profibrotic effects. Altogether, the net effect of PKA appears hard to define. Anyways, your valuable comment inspired us to add a paragraph about GPCR downstream signaling including GPCR kinases and and also referring to PKA (pages 7-8):
„Ang II, ET1 and catecholamines have in common that they exert their function via G-protein coupled receptors (GPCRs) such as the AT1R, ETA, ETA and b-adrenergic receptors. Current pharmacological approaches mainly target these GPCRs. The sensitivity of hormon- induced GPCR activation is modified by G-protein coupled receptor kinases (GRKs) and Regulator of G-Protein Signaling proteins [6]. It has been demonstrated that dysregulation of the GRK isoforms 2-6 correlates with increased blood pressure, poor response to antihypertensive treatment and adverse cardiovascular outcomes [59]. Renal fibrosis could be alleviated by inhibition of the G-protein βγ-subunit in mouse models of cardiorenal syndrome and ischemic acute kidney injury [23]. Further, inhibition of  the downstream kinase PKA may reduce profibrotic signaling and profibrogenic epigenetic priming in diabetic kidney disease [8]. Pharmacological modification of GPCR downstream targets may be a promising approach in hypertensive CKD.“

Comment 3:
The authors should comment on Mohamed et al  J Am Soc Nephrol 27, 745–765 (2016)
Answer:
Thank you very much, as suggested, it was clarified in the text on page 4, that based on data from models of diabetic nephropathy (e.g. Mohamed et al.), IL17 seems to play an ambiguous role in the progression of renal damage.
„Notably, the cytokine IL17 has a differentiated function in diabetic kidney disease: Ablation of intrarenal Th17 cells ameliorated early diabetic nephropathy [25] while administration of low-dose IL17A had beneficial effects [35]. The ambiguity of IL17 signaling in hypertensive kidney disease remains to be elucidated.“

Comments 4-7:
- Ln122 the authors probably mean AT1R blockers
- Ln155 the authors reminder for a REF should be completed
- Ln170 grammar and syntax need addressing, there are few other instances in the manuscript as well
- Ln361 grammar correction
Answer:
We apologize for these mistakes, spelling and wording were corrected.

We hope to find your approval and contentment with our corrections.

Reviewer 3 Report

This is a well writen manuscript and I have no further suggestions. Great work.

Author Response

(The authors gave the same response as above.)

Round 2

Reviewer 2 Report

The authors have made a good attempt to address my comments